# ABO: Abandon Bayer Filter for Adaptive Edge Offloading in Responsive Augmented Reality

## ABSTRACT

Bayer-patterned color filter array (CFA) has been the go-to solution for color image sensors. In augmented reality (AR), although color interpolation (*i.e.*, *demosaicing*) of pre-demosaic RAW images facilitates user-friendly rendering, it creates no benefits in offloaded neural network analytics but only increases the image channels by 3× with higher transmission overheads. Thus, we propose *ABO*, an adaptive RAW frame offloading framework that parallelizes demosaicing with DNN offloading. The contributions are three-fold: First, we design a configurable tile-wise RAW image neural codec to compress frame sizes while sustaining the downstream DNN accuracy under various bandwidth restraints. Second, based on content-aware tiles-in-frame selection and runtime bandwidth estimation, a dynamic transmission controller adaptively calibrates codec configurations to maximize the DNN accuracy under real-time constraints. Third, we further optimize the system pipelining to reduce the end-to-end frame processing latency. Through extensive evaluations on a prototype platform, ABO consistently provides a 40% more frame processing throughput and a 30% less end-to-end latency while improving the offloaded DNN accuracy by up to 15% compared to SOTA baselines. It also presents improved robustness against dim light and motion blur situations.

### ACM Reference Format:
Anonymous Author(s). 2024. ABO: Abandon Bayer Filter for Adaptive Edge Offloading in Responsive Augmented Reality. In *Proceedings of ACM Conference (Conference'17)*. ACM, New York, NY, USA, 13 pages. https://doi.org/10.1145/nnnnnnn.nnnnnnn

## 1 INTRODUCTION

Augmented reality (AR) overlays digital content like images, videos, or sounds onto the real-world environment, enhancing human-environment interactions by adding generated elements. It has extensive applications in surgical assistance [8, 43], E-commerce [31, 53], gaming [4, 39], and education [50, 54]. Accurate perception of physical objects in real-time lays the foundational pillars for AR. Although deep neural networks (DNN) have greatly enhanced machine perception capabilities (*e.g.*, object detection), their computation-intensive nature incurs significant challenges to limited compute resources on mobile AR devices. Thus, edge offloading that transmits data to a nearby edge server for remote DNN analytics has become a leading solution [12, 20].

The human-computer-environment interactions in AR call for low response latency (< 40 ms), high processing throughput (> 25 FPS), and sufficient task accuracy for a *responsive*, *smooth*, and *precise* user experience. The end-to-end response latency denotes the duration from the frame being captured to the DNN inference results being rendered, including frame preprocessing, data compressing (*i.e.*, encoding), two-way transmission, and remote DNN inference steps. To save network transmission, the captured frames are first compressed by an *image codec* before sending, which should effectively reduce image sizes, run efficiently on mobile devices, and dynamically adapt to bandwidth fluctuations. Standard image codecs like JPEG, along with recent offloading approaches [7, 35, 36], all focus on encoding preprocessed RGB images and overlook potential optimizations in image preprocessing. Besides, they are optimized for human browsing experience and could be suboptimal in serving DNN analytics.

Common digital cameras use Bayer-patterned color filter array (CFA) to capture color information into single-channel *RAW frames*, which are then demosaiced to interpret real-world colors into 3-channel *RGB frames*. Demosaicing high-resolution frames can take more than 25 ms on resource-constrained edge devices (*e.g.*, NVIDIA Jetson Nano), occupying a large portion of end-to-end latency since the remaining steps only take below 30 ms in total. We believe demosaicing is only essential to rendering frames into a viewable form but can be skipped for DNN inference. Since RGB images are solely interpolated from RAW ones without any extra input, RAW images should provide a similar task accuracy when feeding into the downstream DNN model despite having less data. We are thereby interested in utilizing RAW frames in edge offloading for DNN analytics. It enables decoupling offloading and demosaicing with no resource contentions (*i.e.*, communication vs. computation), thus the onboard demosaicing overhead can be hidden in parallel pipelines, leading to lower latency and higher throughput.

However, there are three technical challenges in the decoupled pipelines: First, an efficient codec is required to compress RAW images no larger than JPEG files, while ensuring higher downstream DNN accuracy upon decoding. Second, to prioritize real-time interactions, we need to dynamically calibrate codec configurations with accuracy-efficiency tradeoffs in response to runtime bandwidth fluctuations. Third, we need to make both the encoder and the adaptation controller lightweight enough to execute efficiently on edge devices without incurring excessive overhead.

To that end, we propose ABO, an adaptive pre-demosaic RAW frame edge offloading framework for DNN analytics, that decouples demosaicing from the offloading into parallel processes, providing real-time responses under constrained and dynamic network bandwidth. Its design includes three key perspectives. First, we train a configurable RAW image neural codec that operates on sub-frame image tiles with an asymmetric autoencoder (*i.e.*, device-side shallow encoder and server-side deep decoder) to quickly compress

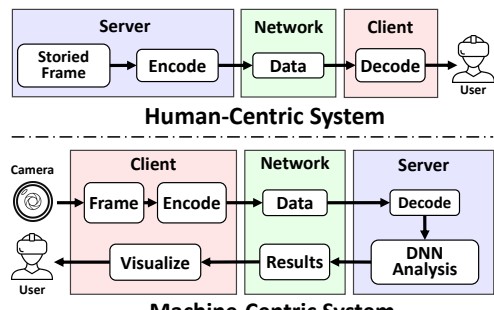

**Figure 1: Comparing two image transmission types.**

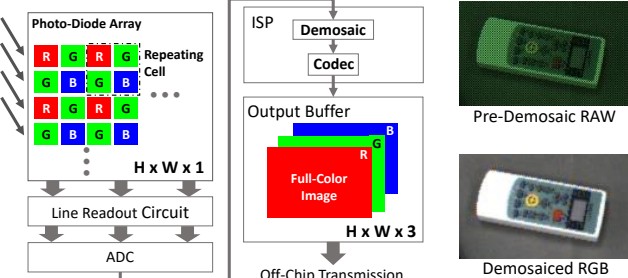

**Figure 2: Typical on-sensor image processing pipeline.**

RAW tiles and utilize server-side computation in exchange for transmission savings. It is trained in a task-aware manner by minimizing both the image reconstruction loss and knowledge distillation loss from downstream DNN, and it is highly configurable in tiles-in-frame selections and the encoded feature dimensions. Second, we design a dynamic transmission controller that adaptively decides the two configuration knobs based on real-time video content and estimated bandwidth, to maximize the downstream task accuracy without violating the real-time latency constraint. Third, we refactor the system pipeline by decoupling and parallelizing demosaicing and offloading processes to optimize bottleneck resource utilizations (i.e., device-side GPU and wireless network).

We extensively evaluate the performance of ABO with datasets collected with a prototype AR system. The results show that ABO outperforms not only JPEG but also the SOTA neural codec baselines, with up to 15% improvement on downstream DNN accuracy while incurring similar bandwidth consumption. Meanwhile, the dynamic offloading pipeline of ABO consistently provides over 40% more throughput and 30% less response latency, achieving the target of real-time experience.

Our main contributions are summarized as follows:

- We are the first to decouple the visual DNN offloading from image demosaicing such that they can run in parallel for higher throughput and lower end-to-end latency.
- We design a tile-based configurable neural codec for RAW images to achieve different latency-accuracy tradeoffs.
- We propose an adaptation controller algorithm to optimize the offloaded DNN task accuracy and frame processing throughput upon network bandwidth fluctuations.
- We implemented both the hardware and software prototype system and performed extensive evaluations to demonstrate the effectiveness and efficiency of ABO.

## 2 BACKGROUND AND MOTIVATIONS

This section reviews preliminary background knowledge on DNN offloading and digital image processing, as well as the main motivations driving this paper.

### 2.1 DNN Offloading for Augmented Reality

DNN analytics (e.g., object detection) are crucial in AR applications as they enhance the way digital content interacts with the real world by helping anchor virtual elements to real-world objects. Due to the limited onboard resources, offloading computation-intensive

DNN analytics to a nearby edge server has been a common practice [33]: The DNN model is loaded on the server to process uploaded frames from the client device (e.g., headset). The detection results are then sent back to the client for user-facing rendering. The offloading pipeline needs to effectively support high-resolution and high-frequency frame processing in real time for an accurate, responsive, and smooth user experience in the augmented world.

To save bandwidth consumption, standard image codecs (e.g., JPEG) are applied on the client to compress image frames before transmission, which actually mismatch with DNN inference needs. As shown in Figure 1, we distinguish conventional *human-centric image transmission* and *machine-centric image transmission*:

- In **human-centric transmission** like content delivery networks (CDN), image frames are transmitted for human viewing after decoding, so standard codecs like JPEG are designed for optimized human perceptual quality.
- In **machine-centric transmission**, the decoded frames are instead processed by DNN models, thus downstream model accuracy should become the new quality metric that guides the neural codec design.

As a solution, the image codec should be refactored to align with the downstream DNN inference and abandon any information unrelated to DNN prediction (i.e., interpolated colors, content intactness). Besides, the human-in-the-loop nature of AR calls for both *low latency* and *high throughput*. Low latency provides responsiveness, while high throughput guarantees smoothness.

### 2.2 Demosaicing in Digital Image Processing

Bayer filter is a fundamental component of digital image sensors. It is technically a CFA placed over the photodiodes of sensors for capturing images. As shown in Figure 2, a typical CFA pattern is a 2x2 repeating unit (RGGB filter cell), and it outputs single-channel *RAW images*. To reconstruct them into the RGB form, demosaicing algorithms are applied [21, 34, 37] to estimate the missing pixel colors based on nearby pixels, which takes 25 to 45 ms on embedded platforms (i.e., Nvidia Jetson Nano). Although demosaiced 3-channel RGB images facilitate human browsing, they do not lead to better DNN inference performance but result in higher transmission overhead. Thus the RAW images, with appropriate codecs, could be better candidates for DNN offloading.

We conduct experiments to validate the hypothesis. Using object detection as the offloaded task, we first train two YOLO models [40] on RAW and RGB images of the same dataset (details in Appendix B).

**Table 1: Accuracy-bandwidth tradeoffs on object detection between different image codecs. YOLOv5 model is used.**

| Image Codec | F1 Score | mAP | Frame Size |
|---|---|---|---|
| RAW | 0.893 | 0.911 | 242 KB |
| ABO-noDistill | 0.873 | 0.878 | 73 ±5 KB |
| ABO-Distill | 0.923 | 0.937 | 75 ±5 KB |
| RGB | 0.888 | 0.904 | 726 KB |
| JPEG | 0.883 | 0.899 | 230±30 KB |

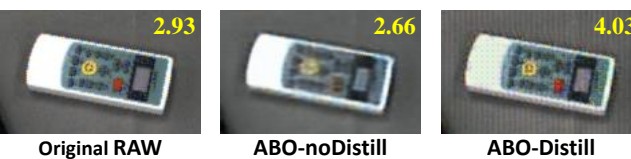

**Original RAW**     **ABO-noDistill**     **ABO-Distill**

**Figure 3: Comparison of the original RAW image, ABO-noDistill decoded image and ABO-Distill decoded image with edge clearness values (Definitions given in Appendix A). Images are demosaiced to RGB formats for better viewing.**

We then evaluate the RGB model on RGB and JPEG-decoded images while evaluating the RAW model on RAW and ABO-decoded images, respectively. The results in Table 1 show that (1) Demosaicing does not enhance the downstream task performance. Instead, the samplings of real-world information remain original and intact in RAW images, which receive better object detection performance than interpolated RGB images (RAW vs. RGB). (2) Without encoding, both frame types lead to excessive frame sizes intolerable for transmission, while ABO and JPEG can effectively reduce the transmitted frame sizes. (3) Although ABO causes a small degradation in model accuracy (ABO-noDistill vs. RAW), finetuning its codec through knowledge distillation achieves even slightly better performance than RAW (ABO-Distill vs. RAW).

We visualize the different image types in Figure 3, and find knowledge distillation achieves targeted image-enhancing artifacts in a mission-oriented manner with better exposed object outlines. It is also noteworthy ABO achieves lower bandwidth consumption than JPEG without sacrificing DNN accuracy, thus presenting a higher potential for offloading.

## 2.3 Efficiency Savings on Pre-Demosaic RAW Frame Offloading

Existing offloading frameworks [26, 28, 51, 52] only optimize the latency from demosaiced RGB images to results rendering but overlook the latency associated with onboard demosaicing. One main motivation of this paper is to decouple *image offloading* and *demosaicing*. Having analyzed its feasibility from the accuracy perspective, here we analyze its efficiency savings.

We compare the standard JPEG-based offloading process with the RAW-based ABO process, which allows onboard demosaicing to run in parallel with transmission and remote DNN analysis. We use an Nvidia Jetson Nano as the edge device and a desktop with Nvidia RTX 4090 GPU as the edge server, whose results are reported in Figure 4. With the decoupled threads, the execution overhead on demosaicing (*i.e.*, 25 ms) is completely hidden behind

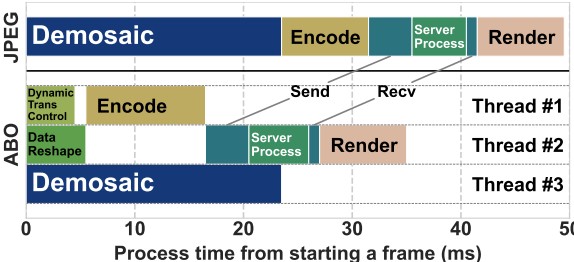

**Figure 4: Frame processing time comparison between *serialized procedure in JPEG* and *pipelined procedure in ABO*.**

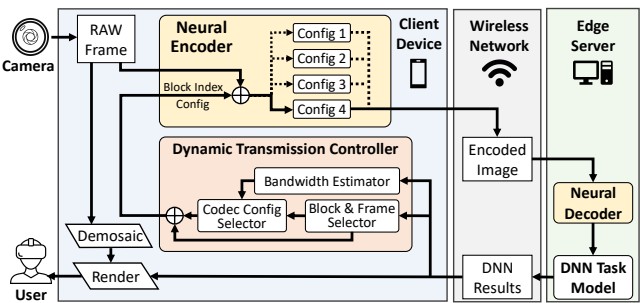

**Figure 5: ABO framework overview.**

image offloading steps. Besides, as indicated in Table 1, single-channel RAW images result in smaller compressed sizes, reducing the network transmission latency. As a result, the end-to-end frame processing latency is reduced from 54 ms to 37 ms, fulfilling the real-time requirement of 40 ms. Its frame throughput increases to over 27 FPS, surpassing the 25 FPS threshold for human viewing.

## 3 ABO FRAMEWORK

We first give an overview of the proposed ABO framework, then introduce its major components with design details.

### 3.1 Overview

An overview of ABO is summarized in Figure 5, which includes two main components: a *tile-wise RAW image neural codec* with different configurations to be calibrated with, and a *dynamic transmission controller* tackling bandwidth fluctuations.

**Tile-wise RAW Image Neural Codec:** It consists of a lightweight encoder on the edge device to compress sub-frame RAW image tiles into feature maps with compressed sizes, and a deep decoder on the edge server to reconstruct the RAW image from transmitted feature maps for DNN inference. Multiple encoding configurations are available to balance the offloaded DNN accuracy and bandwidth consumption during the transmission.

**Dynamic Transmission Controller:** Given a RAW frame, it first selects the tiles that may contain objects, then determines the codec configuration based on the estimated bandwidth and selected image tiles, adapting to dynamically fluctuating networks.

### 3.2 Tile-wise RAW Image Neural Codec

A neural codec is expected to significantly reduce the frame size through encoding while effectively sustaining the downstream

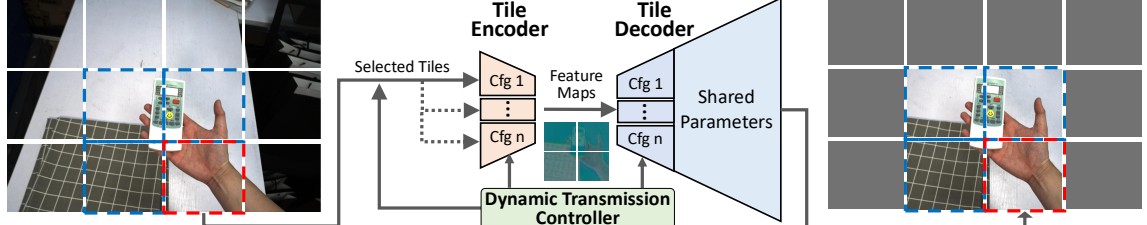

**Figure 6: The tile-wise encoding of ABO with tile-in-frame selection and multiple encoding configurations.**

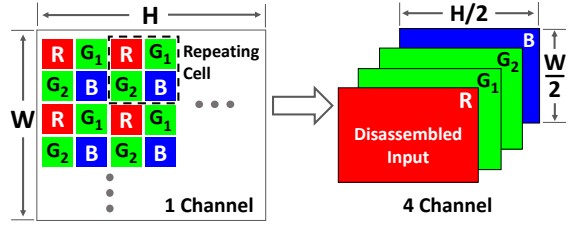

**Figure 7: Color-preserving input preprocessing.**

DNN accuracy on decoded output. ABO's neural codec design is based on two ideas: First, ABO reduces the client-side computation by deploying single-layer neural encoders on edge devices and heavyweight neural decoders on edge servers. Second, instead of encoding a high-resolution RAW image as a whole, ABO separately encodes individual sub-frame tiles and provides multiple encoder configurations, both constituting a tunable space for codec calibration upon bandwidth dynamics.

*3.2.1 Asymmetric Configurable Autoencoder.* The neural codec of ABO is designed to be asymmetric. The encoder only includes a single convolution layer for compressing tile spatial dimensions. The kernel stride and numbers of output channels are different for each configuration, and their parameters are separately assigned. The single convolution layer can run in real-time on edge devices and the existence of multiple encoders only induces reasonable memory overhead. On the other hand, The decoder on the edge server has a much larger scale to secure the reconstruction quality, as it is assumed to be resource-rich. The basic decoder structure is a stack of residual blocks[15] (details in Appendix E). To cope with heterogeneous encoding configurations without hosting multiple deep decoders, as shown in Figure 6, we use a pluggable design: Each encoder configuration has a separate decoder head to unify the feature map dimensions. After that, all configurations share the same decoder layers until the output. All encoder-decoder configurations are trained jointly to enhance their compatibility within the shared decoder parameters.

*3.2.2 Frame Color Preservation.* If a RAW image is fed into the encoder in the original 1-channel format, the grid-patterned RGB information would be erased by the convolution kernel, hard to recover the color information. Instead, we disassemble a RAW image by the color channels in the CFA repeating pattern and stack four disassembled channels to create a 4-channel input (illustrated in Figure 7), so that the mosaiced color information of each channel can be preserved during the encoding-decoding processes. However, though a disassembled 4-channel and the original 1-channel frames

have the same pixel volume and information, the spatial information of a 4-channel input is downsampled by 2× at each dimension which will greatly degrade the extracted high-level spatial features of the DNN model[44]. Thus, the decoded tiles need to be reassembled into the original 1-channel format along the CFA pattern before feeding into the downstream model.

*3.2.3 Tile Partitioning.* Object detection models are highly spatially localized in feature extraction, meaning the detection quality will not be affected by removing unrelated background areas. Furthermore, selectively transmitting subframe areas with rich information helps greatly reduce bandwidth consumption. To achieve this, we evenly split a frame as a grid of $r \times c$ overlapped tiles to preserve the quality of reconstruction from tile padding artifacts. After transmitting the selected tiles with their positional indexes to the edge server, the decoded tiles will be placed at their original position in an empty frame canvas, while the skipped tiles are filled with default pixel colors.

*3.2.4 Learning Objectives.* The neural codec training is divided into two phases. The first phase only focuses on the reconstruction quality by minimizing a mean-squared error (MSE) loss between the original input and the decoded output. In the second phase, we introduce knowledge distillation [17] from the downstream DNN model to enhance the preservation of task-related information, achieving higher accuracy in the downstream task. The frozen downstream model is concatenated to the decoded output and its task loss (*i.e.*, object detection loss) is backpropagated to both the encoder and decoder during their update. Besides, in the distillation phase, losses from different coding configurations are weighted by the proportion of their original MSE losses. Assume there are $m$ train samples in total, and $y_i$ is the codec output of corresponding train sample $x_i$. Then the MSE loss can be described as

$$Loss_{MSE} = \frac{1}{n} \sum_{i=1}^{n} (x_i - y_i)^2. \tag{1}$$

Assume there are $n$ codec configurations in total, and denote the MSE loss of one of the configurations as $\{Loss_{MSE}\}_i$, the weight for this configuration can be described as:

$$w_i = \frac{\sum_{i=1}^{n} \{Loss_{MSE}\}_i}{\{Loss_{MSE}\}_i}. \tag{2}$$

Denote the loss function of downstream DNN visual model as $Loss_{task}$, so that the task loss of configuration $i$ is $\{Loss_{task}\}_i$, and the knowledge distillation loss is:

$$Loss_{KD} = \sum_{i=1}^{n} P_i * \{Loss_{task}\}_i, \tag{3}$$

where $Loss_{task}$ depends on the used downstream model. In this way, all codec configurations have the same loss value at the beginning of knowledge distillation, with the intuition of forcing them to be updated at a similar rate.

## 3.3 Dynamic Transmission Controller

To ensure smoothness and responsiveness against network fluctuations, we design a dynamic transmission controller to continuously calibrate neural codec configurations at runtime. It contains a *codec configuration calibrator* that operates based on an offline profiled look-up table (LUT), a *content-aware tile selector*, and a *lightweight bandwidth estimator*.

*3.3.1 Adaptation Problem Formulation.* The adaptation objective is to maximize offloaded DNN accuracy on each frame satisfying the real-time latency constraint under certain bandwidths. To do so, a LUT is first created offline as $\{keys : [C_i], values : [(P_i, B_i)]\}_{i=1}^{n}$ where $C_i$ is the codec configurations, $P_i$ and $B_i$ are the corresponding profiled task accuracy and bandwidth consumption. $B_i$ is calculated by $B_i = FS_i/T_i^t$ where $FS_i$ and $T_i^t$ are the average tile size and a predetermined transmission time threshold. If the profiled bandwidth consumption of a configuration is higher than the estimated network bandwidth, the transmission time can exceed the threshold leading to system lagging and stuttering. At runtime, a content-aware tile-in-frame selection is performed first, giving a list of selected tiles $LT$, and the numbers of selected tiles $t_s$. Meanwhile, an estimated available bandwidth $EAB$ is given by a lightweight estimator. Then the objective of adaptation can be evolved into finding the codec configuration $C_i$ with the best task accuracy $P_i$ that fits $B_i < EAB$.

*3.3.2 Offline Profiling.* To prepare the aforementioned LUT, we use a small profiling data set from the ABO RAW dataset (details in Appendix B) to measure the average accuracy and tile size. The throughput of codec configurations is obtained by running on live cameras under certain bandwidth constraints without involving the dynamic transmission controller.

*3.3.3 Bandwidth Estimation.* We use a lightweight bandwidth estimator to obtain the estimation of the currently available bandwidth (EAB). The estimation $BW$ is obtained by a simple time differential $BW = \frac{S_{trans}}{t_{end} - t_{start}}$ where $S_{trans}$ is the transmitted bitstream size of the last frame, $t_{start}$ and $t_{end}$ are the staring and ending timestamp of transmission of the last frame.

*3.3.4 Content-Aware Tile Selection.* To enhance content awareness in tile selection, we use detection results from the latest frame as references for tile selection in the next frame. For each new tile, if it overlaps with any object bounding boxes in its previous frame, it will be selected for encoding and transmission. Such a conservative criteria ensures only tiles that are unlikely to contain objects are skipped. However, if previous detection results are inaccurate, the following tile selections could be affected in a cascade. To avoid this case, we use a periodic method to reset potential reference errors in fixed-length time windows. In each window, the first frame is always encoded with all tiles in the highest configuration (called the *key frame*), while the following frames are encoded with tile selection and codec configuration calibration. Since key frame

---

**Algorithm 1:** Runtime Configuration Adaptation.

**Input:** Offline profiled LUT
$LUT = \{keys : [C_i], values : [(P_i, B_i)]\}_{i=1}^{n}$,
inference results of last frame $\{bbox_i\}_{i=1}^{m}$, period
length $l$, estimated bandwidth $EAB$

**Output:** selected tile list $LT$, codec configuration $C_s$

   // Initialization

1  Sort $LUT$ by $P_i$ in descending order;

2  frame_count=1;

   // Main Loop

3  **while** *TRUE* **do**

4     $LT=\{\}$;

5     **if** *frame_count==1* **then**

6        $LT = \{T_i\}_{i=1}^{r \times c}$, $C_s = C_1$;

7     **else**

8        **for** $i \in \{1, 2, \cdots, m\}$ and $t \in \{1, 2, \cdots, r \times c\}$ **do**

9           **if** $T_t \cap bbox_i \neq \emptyset$ and $T_t \notin LT$ **then**

10             append $T_t$ into $LT$;

11        num_tiles = $length(LT)$;

12        **for** $i \in \{1, 2, \cdots, n\}$ **do**

13           **if** $B_i \times num\_tiles \leq EAB$ **then**

14             $C_s = C_i$;

15     frame_count=frame_count+1 **if** frame_count!=l **else** 1;

16     Output $LT, C_s$ to encoder;

17     $\{bbox_i\}_{i=1}^{m}=\{\hat{bbox_i}\}_{i=1}^{k}$;

---

transmissions only happen in low frequencies, their overhead is attenuated across all frames in the window and potential reference error accumulation is upper bounded, thus we guarantee the reliability of tile selection with high throughput and low latency even in high-motion scenarios.

*3.3.5 Adaptation Algorithm.* With the above definitions, the adaptation algorithm can be described as shown in Algorithm 1. For a given frame, if it is the first frame of a controlling window (a key frame), all the tiles will be encoded in the highest configuration and transmitted for DNN inference. The inference results will be stored as a reference. If it is not a key frame, the tile selection will be performed first. For each tile in the frame, if it is not already selected and has a cross-section with any bounding box from the last frame inference results, it will be selected for encoding and transmission. After the tile selection is finished, the number of selected tiles will multiplied by the bandwidth usage profile to obtain the estimated bandwidth request (EBR). The best configuration with an EBR less than the EAB will be used for encoding selected tiles and the DNN inference results will be used for tile selection of the next frame.

## 4 IMPLEMENTATION

### 4.1 Hardware Prototype

As shown in Figure 8, we build a hardware prototype system with a 3D-printed AR goggle modified from Google Cardboard [1]. We use two IMX178 rolling-shutter sensors from SonySemicon [16] as cameras, with a solution of $3072 \times 2048$ and a maximum frame rate of 60 FPS. We use NVIDIA Jetson Nano as the edge device for onboard processing. The edge server is configured with an AMD

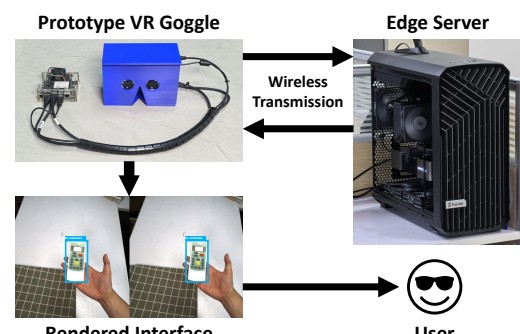

**Figure 8: 3D-printed AR offloading testing platform.**

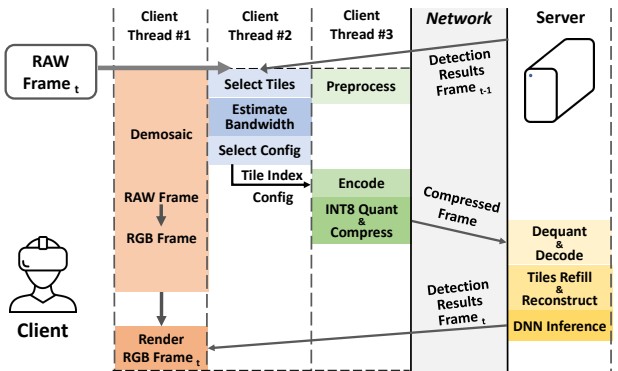

**Figure 9: ABO framework operational workflow.**

Ryzen R9-7950X CPU with a Nvidia RTX4090 GPU and 64GB RAM, comparable to home desktop PCs. The client device and the edge server are connected through a 200 Mbps wireless network.

## 4.2  Software Implementation

We implemented ABO framework with 4500 LoC Python code, using PyTorch 2.0 [38] as the DNN platform. The network connection and data transmission is achieved through the TCP Socket protocol. The downstream object detection model is YOLOv5-7.0. Encoders on the edge device are converted to FP16 TensorRT engines [3]. Lempel-Ziv coding and Huffman coding are employed for data compressing before transmission to further reduce bandwidth consumption.

## 4.3  System Optimizations

*4.3.1  Pipelining.* The pipeline of ABO implementation is illustrated in Figure 9. Multi-threading the demosaic process with the others means the execution time of all the processes except encoding will be extended since the total CPU resource pool is very limited. However, according to our observation, the multi-threaded pipeline still provides 15% more FPS with 17% less end-to-end latency compared to a sequential one.

*4.3.2  Quantization.* We reduce bandwidth consumption by quantizing the encoded feature map into unsigned INT8 using simple *add* and *mul* operations. The parameter of quantize is obtained from the upper and lower bound of encoded feature maps of the training set. The quantization brings only about 0.05% degradation of task performance with a neglectable processing time overhead and a reduction in bandwidth consumption of about 75%.

## 5  EXPERIMENTS

In this section, we commence describing the experimental setups and the datasets employed in our experiments. Then, we provide a comprehensive comparison of overall performance against various baseline methods, followed by robustness evaluation, ablation study, and robustness evaluations.

## 5.1  Experimental Setups

*5.1.1  Dataset:* Since existing public datasets do not contain RAW image frames, we manually collect segments of 10-40 seconds consecutive RAW frames at 30 FPS using the prototype AR device, including different possible real-world AR scenarios. The detailed statistics are summarized in Appendix B.

*5.1.2  Evaluation metrics.* To verify the performance of the proposed framework, we consider the following metrics perspectives.

- **Task Accuracy:** For the object detection task, mean average detection precision (mAP) [13, 27], F1 score, precision, and recall are used to represent the overall task accuracy.
- **Frame Processing Latency:** It is defined as the duration from the frame being captured to the remote detection results being rendered and displayed by the client device.
- **Throughput:** The frame processing throughput is critical to the smoothness of the user experience. It is defined as the number of frames that are processed per second.
- **Bandwidth Consumption:** It measures the average size of transmitted frames, as network bandwidth has become one of the resource bottlenecks in edge offloading.

## 5.2  Compared Baselines

We compare with the following baselines in our experiments:

- **JPEG [47]:** One of the most widely used methods of lossy compression for RGB images, with a scaled compression ratio reflecting the tradeoff between image quality and file size. We set the JPEG configuration with an adaptive encoding quality module according to the available bandwidth.
- **DeepCOD [52]:** A neural offloading framework using different layers of deep compression to achieve efficient transmission while balancing both reconstruction quality and downstream task accuracy.
- **PNC [49]:** An adaptive image offloading framework with a neural encoder-decoder set achieving selectable compression rates via stochastic tail-drop according to the importance of different layers in the feature map.
- **Reducto [26]:** An adaptive frame selection framework with a per-frame differential extractor on the client and offline profiles for scenarios on the server to control which frames to transmit within a stream.

## 5.3  Offline Accuracy Profile

An optimal neural codec should reduce the frame into smaller sizes while sustaining higher downstream task accuracy. We therefore measure the accuracy and bandwidth tradeoffs between different configurations in offline profiling, which represents the Pareto boundaries of the compared frameworks. Specifically, for each configuration, we measure the task accuracy metrics after decoding

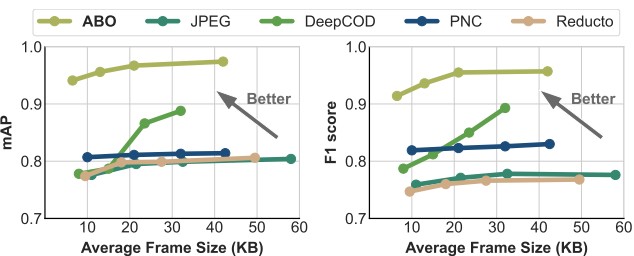

Figure 10: Accuracy and frame size profiles.

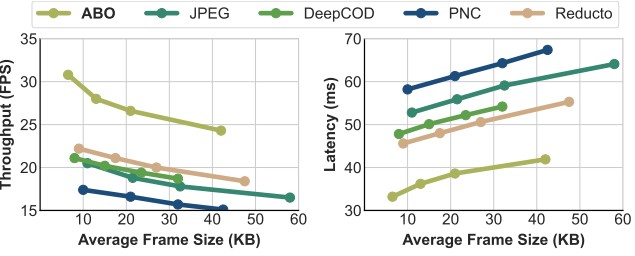

Figure 11: Throughput & latency under 20 Mbps bandwidth.

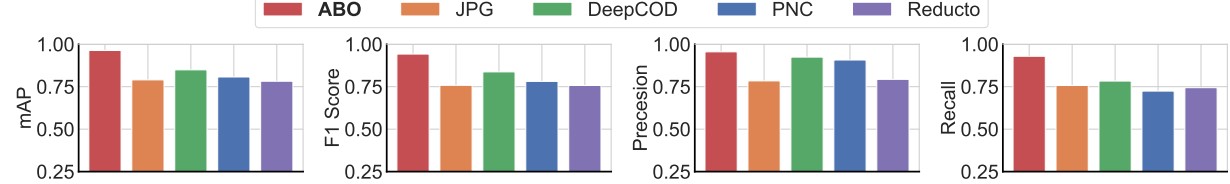

Figure 12: Adaptation accuracy under bandwidth dynamics.

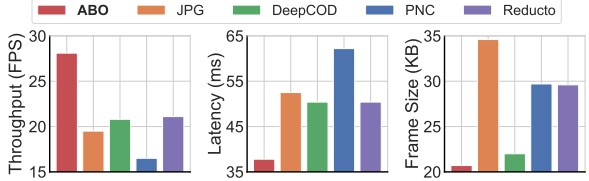

Figure 13: Adaptation efficiency under bandwidth dynamics.

and the average compressed frame sizes across the test segments as shown in Figure 10. Each line represents a framework and each point represents a concrete configuration. ABO reaches the highest Pareto limit among all frameworks across all configurations. Compared with the JPEG pipeline, ABO has an over 15-20% better task accuracy while only using 30-50% less bandwidth, demonstrating the importance of task-aware end-to-end code training in DNN offloading. Besides, although based on the same model architecture, ABO outperforms the frame-level encoding of DeepCOD significantly benefiting from its tile-wise encoding and object-free tile filtering during encoding. Finally, the sole frame selection in Reducto presents poor bandwidth efficiency in downstream tasks.

## 5.4 Offline Throughput and Latency Profile

The end-to-end frame processing latency and the throughput under limited bandwidth are directly related to the responsiveness and smoothness of user experience. Therefore, we upper bound the wireless network to 20 Mbps and measure the above two metrics for different configurations within each framework, using all collected test segments. As illustrated in Figure 11, under the same frame sizes, ABO achieves the highest throughput and the lowest end-to-end latency, being the only framework surpassing the 25 FPS threshold of real-time inference, demonstrating that the advantages of ABO also come from its higher client-side computation efficiency and system pipelining. Compared to the JPEG pipeline, ABO increases the throughput by 50% while reducing the end-to-end latency by 35%, highlighting the savings by parallel demosaicing and DNN offloading. PNC struggles with 15% lower throughput and 10% higher latency than JPEG, despite its channel-dropping

effort, because its encoder still has a large scale that leads to high computation overhead. Other baselines have only improved 3% to 15% of throughput and 5% to 10% of end-to-end latency compared to JPEG without refactoring the underlying codec.

## 5.5 Adaptation Performance

We further perform end-to-end comparisons when individual configurations are integrated into an adaptive framework under network dynamics. To conduct the adaptation experiments in a reproducible way, we randomly generate network bandwidth traces (as summarized in Appendix C) and replay the network traces to each video segment and each framework, respectively. The network bandwidth is dynamically bounded using Linux Traffic Control (tc) [2]. The results are shown in Figure 12 and Figure 13. ABO consistently outperforms the baselines in both downstream task accuracy and frame processing throughput. Neural encoding frameworks generally achieve better task performance (ABO, DeepCOD, and PNC), where ABO demonstrates the best efficiency and accuracy tradeoff. DeepCOD and Reducto deliver increased throughput than JPG but still can not meet the smooth browsing requirement (i.e., > 25 FPS). PNC suffers from long encoding time although the transmitted frame sizes are compressed through its channel-dropping strategy. In AR applications, since objects are mostly in high motion to the user, the frame selection in Reducto leads to poor downstream accuracy. Finally, ABO also achieves the lowest end-to-end latency, thanks to its demosaic-free offloading pipeline. In summary, the results prove that ABO provides the highest quality with guaranteed responsiveness and smoothness to the users.

## 5.6 Environmental Robustness

Experiments in this subsection evaluate two challenging scenarios (low-light and high-motion, details explained in Appendix D) in AR applications that require higher robustness.

*5.6.1 Low-Light Scenario.* We manually change the luminosity scale of the collected video frames from 1 to 1/2 and 1/4, and compare the accuracy degradation between ABO and JPEG codec on

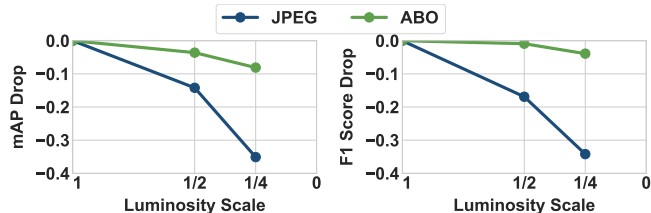

Figure 14: Accuracy degradation under low luminosity.

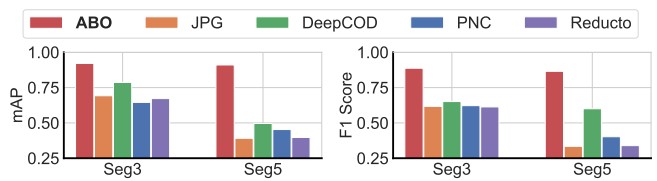

Figure 15: Adaptation performance in high-motion videos.

their highest configurations. As shown in Figure 14, when operating in low-light conditions, ABO achieves lower accuracy degradation below 0.1 while JPEG has an unbearable 0.35 at 1/4 luminosity.

*5.6.2 High-Motion Scenario.* Active user interactions can make high-motion scenarios common in AR. Therefore, we separately analyze the downstream accuracy of two video frames (*i.e.*, Seg3 and Seg5 ) that were collected with high camera motions (efficiency shown in Appendix D). As shown in Figure 15, neural codecs tend to have better robustness than frameworks using standard JPEG encoding. Besides, ABO outperforms the baselines with a larger margin than overall evaluations in Figure 12, being the only framework to sustain over 0.85 mAP in both videos.

## 5.7 Ablation Study

To inspect the optimization brought by each module, we designed two ablation experiments. One removes the distillation process in the offline codec training, while the other one removes the tile selection module in dynamic transmission control. The results are shown in Table 2. Knowledge distillation does not affect frame size, throughput, or end-to-end latency, but brings a huge improvement in task accuracy (over 5% mAP), making it an essential component in ABO. Besides, tile selection saves 23% in average frame sizes with on average 29% tiles dropped, hence the improvement in throughput and latency, without degrading the downstream accuracy.

## 5.8 Overhead Quantification

We quantify the memory usage and energy overhead of ABO to evaluate its applicability in mobile AR devices. We monitor the power and memory usage throughout the testing periods. The Power consumption is measured with a Monsoon HV power moniter [19] at 0.2 ms intervals. The cumulative overhead distribution curves of JPEG and ABO are visualized in Figure 16 In power consumption, the average power of ABO is 10 W, which increases by < 10% than 9.5 W of JPEG pipeline, demonstrating its high power efficiency. In memory usage, after utilization, ABO consumes 1080 MB of memory while JPEG consumes 340 MB. This is reasonable since we host multiple neural encoders on the edge device, but its absolute memory consumption is acceptable to current AR device capacities.

Table 2: ABO ablation study results.

| | F1 Score | mAP | Avg Frame Size | Throughput | Latency |
|---|---|---|---|---|---|
| ABO | 0.941 | 0.963 | 19.6 KB | 29.0 FPS | 37.1 ms |
| ABO-noDistill | 0.875 | 0.907 | 20.4 KB | 28.9 FPS | 37.3 ms |
| ABO-allTiles | 0.942 | 0.963 | 25.4 KB | 27.2 FPS | 39.2 ms |

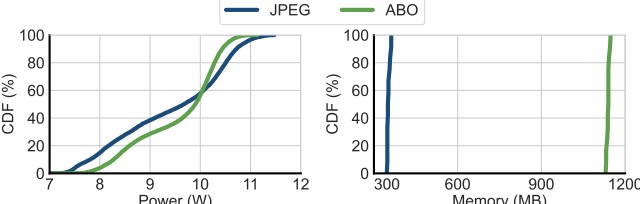

Figure 16: Energy and memory overhead quantification.

## 6 RELATED WORK

Since the emergence of CCD image sensors, Bayer-patterned CFA has been the go-to solution for creating digital full-color images. Besides, encoding images with DNN models has been rapidly developed in recent years [29, 45, 46]. However, despite several works have encoded CFA raw images [5, 9, 11, 25, 42], there is still a long way to using neural codecs for CFA images on mobile devices with insufficient computing power. Traditional DNN methods targeting to improve the the shear performance (such as PSNR and SSIM) and human-view-experience use more parameters in the DNN codec, means that the time overhead and performance consumption are too much, making it hard to have any practical utilization in mobile computing scenario. Hardware methods have also been considered [32], though the performance is outstanding, a custom-made sensor module is too costly for low-end devices. There are also attempts to use Bayer image for object detection task [30], but it requires a distinct model that only targets a single mission.

The offloading technique is proposed to address the restraint of insufficient computing power of edge devices [33]. Using such technique in AR/VR scenarios has been a new investigation direction [6, 14, 18, 23, 24]. Among existing solutions, there are several frameworks considering the rising trend of using high-resolution to improve the quality of user experience. [41, 48, 55] However, there is no investigation into using pre-debayered RAW images in mobile offloading scenarios, expect ABO.

## 7 CONCLUSION

In this paper, we introduced ABO, an adaptive RAW frame offloading framework for DNN analytics in AR applications. It relies on three main designs: First, it decouples demosaicing and offloading into parallel processes through system pipelining for reduced latency; Second, it contains a tile-wise RAW image neural codec with multiple configurations; Finally, it adaptively calibrates the coding configurations based on the content-aware tile selection and runtime bandwidth. Through evaluations on a prototyped hardware platform, ABO constantly achieves up to 15% improvement in downstream task accuracy while increasing the frame processing throughput by 40% and reducing the end-to-end latency by 30% with similar bandwidth consumption, compared to SOTA baselines.

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

## A SOBEL-TENENGRAD EDGE CLEARNESS VALUE

The edge clearness value is calculated by the Sobel-tenengrad operator, which is commonly used to assess the sharpness or clarity of image edges. The Sobel operator is an edge detection method used to calculate the gradient of the image intensity in both the horizontal (x) and vertical (y) directions. The two convolution kernels (horizontal gradient $G_x$, vertical gradient $G_y$) for the Sobel operator are defined as follows:

$$G_x = \begin{bmatrix} -1 & 0 & 1 \\ -2 & 0 & 2 \\ -1 & 0 & 1 \end{bmatrix} \quad (4)$$

$$G_y = \begin{bmatrix} -1 & -2 & -1 \\ 0 & 0 & 0 \\ 1 & 2 & 1 \end{bmatrix} \quad (5)$$

The gradient magnitude at each pixel is then calculated using the following equation:

$$G = \sqrt{G_x^2 + G_y^2} \quad (6)$$

This magnitude represents the edge strength at each pixel. Larger values indicate stronger edges, which typically correspond to sharper image features. Then we can calculate the Tenengrad focus measure $T$, which is used to quantify the overall sharpness of the image, by summing the squared gradient magnitudes across the entire image. Higher values of $T$ indicate a clear or sharper image, as they reflect stronger edges throughout the image. The formula is given by:

$$T = \sum_{i,j} G(i,j)^2 \quad (7)$$

## B DATASET DETAILS

In our experiments, the downstream vision task deployed on the server is YOLO-based object detection. Existing common datasets [10, 27] are all collected and labeled with RGB standards like JPEG, and simulating RAW images from such data is nearly impossible. On the other hand, existing RAW datasets [5, 11, 22] either simply do not have the sufficient amount of images to create a persuading test environment, or are not designed for vision tasks but only compressive encoding. Thus, for experiments, we construct a new RAW dataset, the ABO RAW dataset, with 17 types of common objects in campus offices, containing over 10,000 images with high-quality labeled bounding boxes. All the images are captured with a resolution of 3072*2048 under 8-bit non-packing Bayer mode with the digital gain set to zero to reduce any possible noises and secure image quality. This dataset is designed not only to serve the object detection task but also to improve the robustness under common challenges in AR/VR scenes like dim light or motion blur. To make the proxy of AR/VR scenario more viable and improve the robustness, random movements of the camera and objects are introduced during the data collection process to improve the tolerance of motion blur in real-life AR/VR scenarios. Roughly 5% of the total dataset is collected under insufficient luminosity conditions to robustness under indoor environments. We also introduced several frames of pure black and white for gamma calibration.

However, the dataset is collected one frame at a time to maintain the variety of samples, meaning it cannot be used as frame streams

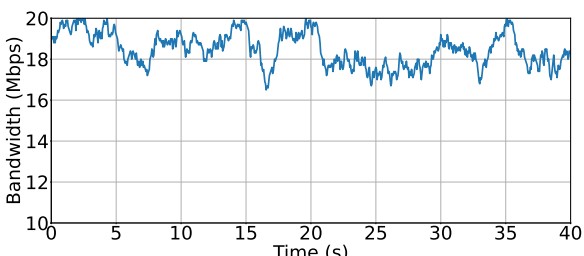

**Figure 17: Bandwidth Trace 1.**

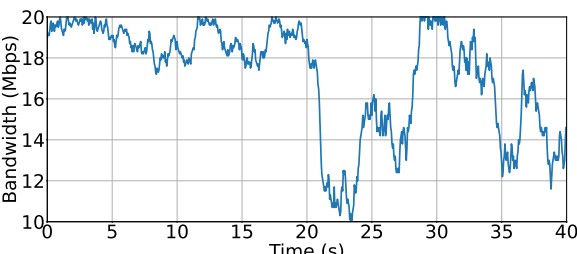

**Figure 18: Bandwidth Trace 2.**

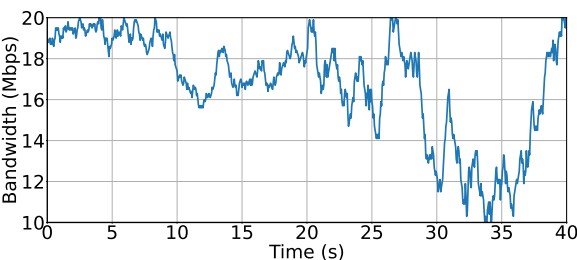

**Figure 19: Bandwidth Trace 3.**

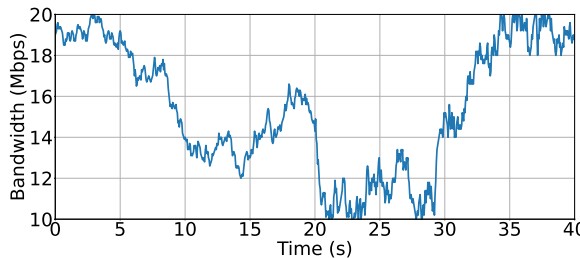

**Figure 20: Bandwidth Trace 4.**

since the frames are not consecutive. To evaluate the framework, we also collected 5 segments of consecutive frames at a fixed frame rate of 30 FPS using the prototype AR device. The segments include different possible real-world AR/VR scenarios within time lengths various from 10 to 40 seconds, simulating both short-burst and long-term services.

## C GENERATED BANDWIDTH TRACE

The bandwidth traces are generated by a binomial random algorithm. The traces are applied to the WiFi chip through a shell script using Linux Traffic Control (tc). The script is started along with the

Figure 21: The configurable image neural codec in ABO.

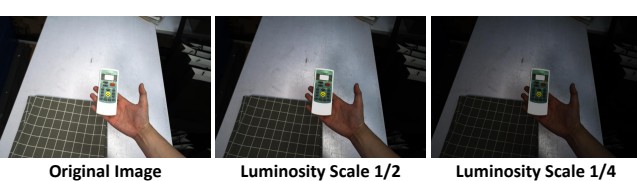

**Original Image**     **Luminosity Scale 1/2**     **Luminosity Scale 1/4**

Figure 22: Example of Low-Light Frames

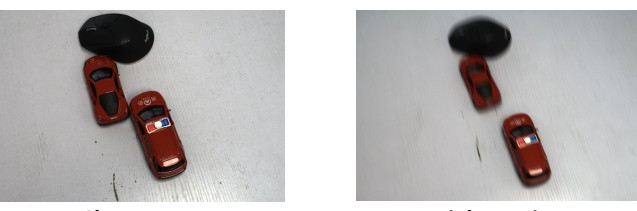

**Clear Image**     **High-Motion**

Figure 23: Example of High-Motion Frames

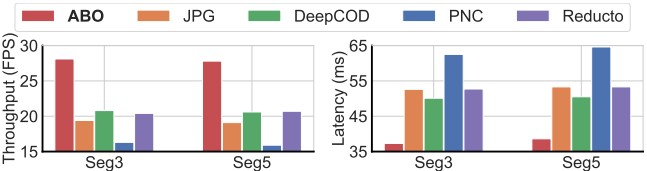

Figure 24: Efficiency under high-motion

main Python script. But since the tc rules need to be erased before the next utilization, meaning an apply-reset-apply cycle is longer than the time granularity of which when the trace is generated, the tc controlling script can only sample from the trace at the corresponding time. The traces are illustrated in Figure 17, Figure 18, Figure 19, and Figure 20.

## D DETAILS OF CAPABILITY EXPERIMENTS

Here we present the visualization of low-light frames (Figure 22) and high-motion ones (Figure 23). In Figure 24, we present the efficiency of ABO under two high-motion test segments. The results show that under the pressure of high-motion frames, ABO not only achieves high task accuracy but also consistently provides the highest throughput and the lowest latency, achieving a smooth and responsive experience.

## E DETAILED STRUCTURE OF ABO NEURAL CODEC

The detailed model structure of ABO neural codec is illustrated in Figure 21. There are 4 configurations in total, decided by convolution kernel strides (2 and 4) and output channels (4 and 8) of the encoder. Upon being encoded and transmitted, the feature map will be divided into two types according to the stride and put into different decoder heads. The feature maps encoded with stride 2 will go through a single convolution layer, while the ones encoded with stride 4 will go through a ResBlock to be upscaled. At this point, the dimensions of the two types of feature maps are roughly aligned. They will go through two different padding layers to be exactly aligned on all dimensions after being output by the same ResBlock. After being further processed by other layers, the feature maps are finally decoded and ready to serve as input for the downstream DNN model.

## F LIMITATIONS AND FUTURE WORK

Here we briefly discuss the existing limitations we identified on ABO that can be potential future directions for extension.

**Pretraining Overhead:** Though the neural codec can be universally utilized for all kinds of data, the downstream DNN model still needs to be trained with RAW frames of required tasks since the input channel has changed. There is also some room for improvement in ABO. The possibility of tile-level codec configuration calibration remains, meaning that the bandwidth consumption can be further reduced. There is also the potential for using a scene-targeting online calibration module to further optimize the performance of the tile-selection module.

**Limited RAW Image Data:** The current ABO design relies on large-scale RAW image datasets for pretraining. Although we have

made a significant amount of effort to collect data simulating daily AR scenarios, its overall scale is still limited compared to standard image benchmarks like ImageNet [10] and COCO [27]. It drives us to think if we can use a large-scale RGB image dataset to pretrain the autoencoder (*i.e.*, extract the semantic feature patterns) and adapt it to RAW input with limited RAW images (*i.e.*, learn to deal with single-channel input), so the data challenges can be greatly alleviated.

**Subframe Codec Calibration:** Current design of ABO transmission controller only calibrates codec configuration on frame-level. There is a potential for using a scene-aware module to calibrate codec configuration on a tile level, thus the bandwidth consumption can be further reduced.

**Multi-Modal Fusion:** The great success of augmented reality not only comes from visual DNN analysis but also from close collaborations between multi-modal fusion (*e.g.*, image, audio, video) that creates an immersive experience for users. However, the current ABO design only considers image data as the input but ignores the potential optimizations that we can perform within multi-modal data streams. We believe the general philosophy of decoupling the edge offloading pipeline from any unnecessary preprocessing steps could fit into various data formats, with corresponding signal processing knowledge.

**Multi-Task Compatibility:** We have used object detection as the only downstream DNN task in our experiments. However, in practice, multiple DNN models can be simultaneously applied to analyze the offloaded frames, serving different application purposes. How to make sure their compatibility within the knowledge distillation of ABO pretraining could be a challenging problem. Within this context, we believe a self-supervised learning paradigm (*e.g.*, contrastive learning or masked autoencoder) that seeks to learn general data semantics without task information could be a promising solution to enhance the generalizability of offloaded RAW frames to heterogeneous tasks.

