# OpenReview forum: "ABO: Abandon Bayer Filter for Adaptive Edge Offloading in Responsive Augmented Reality"
_ACM.org/TheWebConf/2025/Conference — WWW 2025 Oral_

### Official Review · Reviewer_SoDp · 2024-11-25

**Novelty:** 5
**Technical Quality:** 6

**Review:**

**Overview:**
This paper presents a novel system framework named ABO, which modifies the pipeline of AR edge offloading. The results demonstrate significant improvements in both accuracy and latency, with some limitations and problem of the motivations though. Overall, this paper makes substantial technical contributions and provides extensive evaluations, advancing the study in this field.

---

**Strengths:**

- **Observations:** The paper offers well-detailed and coherent observations from both technical and writing perspectives, effectively motivating the study.
- **Writing and Layout:** The clear writing and well-structured layout, including figures and tables, make this work accessible and easy to understand.
- **Novel Pipeline Design:** The decoupling of the demosaic process from the offloading pipeline allows for parallel processing, significantly reducing latency with only a minor, acceptable increase in computing resources.
- **Adaptation:** The design adeptly balances accuracy and efficiency, showcasing good adaptability.

**Cons and Limitations:**

- **Related Work:** The paper lacks sufficient technical details about existing baselines. Specifically, it should elaborate on how ABO differs from previous approaches. Additionally, background information on auto-encoder and detection models should be provided to give readers a comprehensive overview, especially since the positive results largely stem from these areas.
- **Motivation:** It’s important to address why software preprocessing is utilized over ISP hardware, which is typically very fast for image preprocessing in many mobile devices. Clarification on why AR devices do not leverage ISPs in your setup would strengthen the motivation. The device in your setup, NVIDIA Jetson Nano, not common for commercial AR equipments. We can not assume they do not have ISPs.
- **Robustness:** The evaluation of robustness is limited to low-light and fast-motion scenarios.
    - However, complex background scenarios could present more relevant challenges, particularly due to potential issues with your tile-selection mechanism.
    - Furthermore, the connection between low-light conditions and your approach requires further explanation.
    - Your fixed-reset strategy might not be adequate across varied scenarios, such as those with complex backgrounds, which could lead to poor predictions and incorrect tile selections. An adaptive reset strategy may be necessary.
- **Task-Aware Streaming/Transmission:**
    - While you highlight the benefits of parallel rendering and offloading, the task-oriented compression requires further evaluation against other task-oriented communication methods.
    - The related work section should clarify existing work and how your method compares, particularly in relation to semantic communication, which focuses on transmitting semantics rather than raw data. Relevant references include:
        [1] https://arxiv.org/abs/2402.01064)
        [2] https://ieeexplore.ieee.org/abstract/document/9953076
- **Codec Generalisability:**
    - As noted, the downstream tasks are currently limited to object detection, suggesting the need to retrain the auto-encoder for other tasks. In practical scenarios, multiple downstream tasks (e.g., segmentation, reasoning) may be required, potentially leading to compression degradation.
    - It’s important to address whether codecs need to be re-trained for various task combinations, such as object detection plus segmentation, or object detection plus classification. How to tackle the complexity of these un-certain task types?
- **Dataset for Evaluation:**
    - While the collected data is sufficient for demonstrating system feasibility, further verification on more extensive datasets is necessary to ensure that the codec and system perform well in diverse, real-world scenarios.
    - The limited dataset raises concerns about **domain adaptation**, as the codec may not generalise well to broader applications. For example, how does the codec behave on other sensor models, devices, picture resolutions. Again, we emphasise the generalisation is the key concern for system designs.
- **Bandwidth Dynamic:** use some real-world dynamic data could be more convincing.
- **Minor ones**:
    - fig1, how do you define human-centric? I can not tell much thing about human-centric
    - could you cite/back when you claim 40ms/frame is a requirement for human viewing?
    - line 572 starting typo
    - figure10,11 colours are too close, hard to distinguish
    - line 1058 except?
    - fig3 figures are not horizontal

**Questions:**

1. **Software vs. Hardware Preprocessing:** While the parallel preprocessing of images is commendable, could you elaborate on why software-based CFA and demosaic processes are employed? To my understanding, most mobile devices are equipped with hardware ISPs capable of real-time preprocessing for digital imaging. How does your approach benefit from using software in this context, especially considering the efficiency of hardware ISPs?
2. **Tile Partitioning:** Could you provide more detail on how you determine the size of the r \times c tile partitions? What criteria or considerations guide this decision, and how does it impact the system’s performance?
3. **Encoder-Decoder-Detection Process:**
    - **Direct Use of Feature Maps:** It seems more intuitive to use the feature maps directly for detection. What reasons do you have for reconstructing the feature map instead of feeding it directly into the detection model, particularly when the detection model is custom-trained for your setup?
    - **Color Preservation:** How critical is color preservation in your approach? Does maintaining color integrity significantly impact detection accuracy, or could this be an area for optimization?
4. **Learning Objectives:** The formulation of your learning objectives is not entirely clear. Could you provide a more detailed explanation of this aspect?

**Reviewer Confidence:**

3: The reviewer is confident but not certain that the evaluation is correct

**Scope:**

3: The work is somewhat relevant to the Web and to the track, and is of narrow interest to a sub-community

---

### Official Review · Reviewer_FfaF · 2024-11-30

**Novelty:** 5
**Technical Quality:** 6

**Review:**

This paper proposes an adaptive RAW frame offloading framework, called ABO, based on an autoencoder and a dynamic transmission algorithm to reduce resource consumption and improve the performance of downstream tasks. The paper starts with some measurement experiments and introduces some background knowledge to explain the motivation. Then, the paper describes the framework design of ABO, which mainly includes a raw image neural codec based on an autoencoder and a dynamic transmission controller. Finally, the experiments show that the proposed framework can improve the accuracy of the downstream task while increasing throughput and reducing latency.

The motivation of this paper is clearly explained and the author states that it is the first approach to decouple visual DNN offloading from image demosaicing. The paper also builds a prototype and dataset to conduct the experiment. However, it would be better to give a little more description of the design idea and framework detail.

**Questions:**

1. In section 2.2, it states that without encoding, RAW and RGB lead to excessive frame size intolerable for transmission. However, the frame size of RAW (242KB) is similar to JPEG (230KB) in table 1. It is believed that their transmission costs may also be similar.
2. In section 2.3, the ABO achieves lower latency with multi-process pipelines. However, if we apply the pipeline design in JEPG as a 2-stage pipeline including demosaic and encoding, it can also reduce the latency. It would be more convincing to add a little discussion to this question.
3. What is the definition of $P_i$ in equation (3)? If $P_i$ is the task accuracy, it is strange to multiply the task accuracy and the task loss and then use it as the knowledge distillation loss, because the task loss is usually related to the accuracy.
4. In the content-aware tile selection algorithm, how does the algorithm get the first object detection result at the beginning of a time window? In figure 9, it looks like the results come from the downstream task and it couples with the downstream task. So how does the algorithm handle situations where the downstream task changes?
5. Previous work has provided some real-world network traces. Why does the paper choose to use randomly generated bandwidth data in the experiments?

**Reviewer Confidence:**

4: The reviewer is certain that the evaluation is correct and very familiar with the relevant literature

**Scope:**

4: The work is relevant to the Web and to the track, and is of broad interest to the community

---

### Official Review · Reviewer_hAF4 · 2024-12-02

**Novelty:** 4
**Technical Quality:** 5

**Review:**

# Paper Summary
This paper proposes a new adaptive RAW frame offloading framework to accelerate the demosaicing process. The proposed framework consists of three main contributions: a configurable tile-wise RAW image neural codec, a dynamic transmission controller, and an optimized system pipeline. Experimental results demonstrate the advantage of the proposed approach.


# Pros

- This paper utilizes DNN to deal with an interesting task on the edge device, demosaicing. The proposed approach utilizes a lighweight DNN model to substitute the traditional image processing approach.

- The framework includes a controller that can adjust the codec configuration based on the bandwidth and computation requirements.

- The authors build a hardware prototype to evaluate the on-device performance.


# Cons

- The topic seems a little bit off-topic for the Web Conference. The main contribution is to propose a new approach for AR devices, which is more suitable for mobile conferences or VR/AR conferences. This is an major concern as in the references, there are no papers from the WWW or Web Conferences.

- Although the authors report latency on real devices, they didn't report two other important metrics for mobile devices: the energy consumption and memory cost. These two metrics are also important for tasks running on mobile devices.

**Questions:**

- Why is the paper relevant to WWW conference? Can you explain maybe which topic in the CFP does the paper match?

- How is the energy and memory cost of the DNN-based approach? Is the overhead acceptable?

**Reviewer Confidence:**

3: The reviewer is confident but not certain that the evaluation is correct

**Scope:**

2: The connection to the Web is incidental, e.g., use of Web data or API

---

### Official Review · Reviewer_tjav · 2024-12-02

**Novelty:** 5
**Technical Quality:** 6

**Review:**

This paper proposes ABO, an adaptive offloading framework for AR based on raw frames. By segmenting raw frames and performing adaptive neural compression, ABO is able to accelerate AR applications and reduce bandwidth consumption. The idea of performing downstream tasks directly on raw frames was the motivation and main innovation of ABO, and this idea was interesting and attractive. The experimental section also provides strong evidence that ABO has good performance. This paper is clear and easy to understand.

**Questions:**

I do like ABO, but there are still few minor questions.
1. regarding experiments, can you show a breakdown comparison between ABO and other baselines in terms of fps and latency, so that it's easier to understand how ABO is optimized? For example, ABO requires Ams for decoding and Bms for downstream tasks, so the total execution time A+Bms, hence fps is 1000/(A+B).
2. Still on the subject of experimentation, Figures 14 and 16 only provide results for jpegs while the others provide full baseline results. Is there a reason for this? Also, in terms of baselines, has there been any consideration of comparison with video based coding solutions?
3. In the Online Adaptive Algorithm section, it can be seen that each tile is assigned the same quality. Has unequal quality allocation been considered? Also, is the EAB estimate slightly rough, and how do you ensure that violations of the latency constraints are avoided as much as possible? Also, how is r*c determined? Finally, should a 'break' be included in line 14 of Algorithm 1 to ensure that the chosen codec configuration is the one with the highest task accuracy among the configurations with lower bandwidth consumption than EAB.

**Reviewer Confidence:**

4: The reviewer is certain that the evaluation is correct and very familiar with the relevant literature

**Scope:**

4: The work is relevant to the Web and to the track, and is of broad interest to the community